# Soluble Urokinase Plasminogen Activator Receptor (suPAR) in Autoimmune Rheumatic and Non Rheumatic Diseases

**DOI:** 10.3390/jpm13040688

**Published:** 2023-04-19

**Authors:** Mariangela Manfredi, Lieve Van Hoovels, Maurizio Benucci, Riccardo De Luca, Carmela Coccia, Pamela Bernardini, Edda Russo, Amedeo Amedei, Serena Guiducci, Valentina Grossi, Xavier Bossuyt, Carlo Perricone, Maria Infantino

**Affiliations:** 1Immunology and Allergology Laboratory Unit, S. Giovanni di Dio Hospital, 50143 Florence, Italy; mariangela.manfredi@uslcentro.toscana.it (M.M.);; 2Department of Microbiology, Immunology and Transplantation, University of Leuven, 3000 Leuven, Belgium; 3Department of Laboratory Medicine, OLV Hospital, 9300 Aalst, Belgium; 4Rheumatology Unit, S. Giovanni di Dio Hospital, 50143 Florence, Italy; 5Department of Clinical and Experimental Medicine, University of Florence, 50139 Florence, Italy; 6Department of Laboratory Medicine, University Hospital Leuven, 3000 Leuven, Belgium; 7Rheumatology Unit, Department of Medicine and Surgery, University of Perugia, 06122 Perugia, Italy

**Keywords:** suPAR, rheumatoid arthritis, autoimmune diseases, rheumatic diseases

## Abstract

The soluble urokinase plasminogen activator receptor (suPAR) is the bioactive form of uPAR, a membrane-bound glycoprotein, and it is primarily expressed on the surface of immunologically active cells. Mirroring local inflammation and immune activation, suPAR has gained interest as a potential prognostic biomarker in several inflammatory diseases. Indeed, in many diseases, including cancer, diabetes, cardiovascular diseases, kidney diseases, and inflammatory disorders, higher suPAR concentrations have been associated with disease severity, disease relapse, and mortality. Our review describes and discusses the supporting literature concerning the promising role of suPAR as a biomarker in different autoimmune rheumatic and non-rheumatic diseases.

## 1. Introduction

The urokinase plasminogen activator receptor (uPAR), also referred to as urokinase and vitronectin, is primarily expressed on the surface of immune cells, such as macrophages, monocytes, activated T-lymphocytes, neutrophils, but it is also found on fibroblasts, endothelial cells, megakaryocytes, vascular smooth muscle cells keratinocytes and some cancer cells [1,2,3,4,5]. uPAR, the soluble form of the membrane-bound receptor uPAR, is bound to the membrane by a glycosyl posphatidylinositol (GPI) anchor. Soluble uPAR (suPAR) is produced by proteolytic cleavage in the linker region. suPAR is present in blood or other biological fluids, such as saliva, cerebrospinal fluid, and urine. It was first discovered by Danish researchers in 1990 as a biomarker involved in cancer and its progression; later, it was associated with patients’ prognoses of bacterial and other infections, opening the door for its investigation as a biomarker in sepsis [6]. Indeed, the expression of uPAR and suPAR is increased during inflammatory states [1] and, for this reason, elevated suPAR concentrations have been associated with disease severity/relapse and mortality in various immunological diseases [1].

From a molecular perspective, uPAR consists of three domains (D1–D3) that are connected by a linking region between D1 and D2–D3. The linker region and the GPI anchorinclude cleavage sites, and the three main suPAR isoforms are full-length suPAR I-III, suPAR I, and suPAR II-III. An “SRSRY” sequence that is involved in chemotaxis is revealed when uPAR/suPAR is cleaved in the linker region (Figure 1). On the other hand, suPAR, the bioactive form of uPAR [2], has a secondary structure of 17 anti-parallel β-sheets with three short α-helices. suPAR belongs to the plasminogen activator (PA) system, a group of proteases and protease inhibitors regulating the activation of the zymogen plasminogen into its proteolytically active form, plasmin [7]. Notably, the plasminogen is a common plasma protein that can be found in different zymogenic forms, and the plasmin is widely known for its role in blood clot breakdown (fibrinolysis) [8]. In detail, the tissue-type PA (tPA) and urokinase PA (uPA) are the two types of extracellular proteolytic enzyme structures known as the PA system, expressed during many different cellular events [5]. The main component of the PA system, the uPA family, is involved in pro-inflammatory processes such as tissue remodeling [9,10], chemotaxis promotion as well as cancer progression and metastasis. Under normal physiological conditions, uPA performs its function in the tPA presence [11]. Activated uPA-uPAR releases suPAR that regulates some proteolytic cascades, cleaving plasminogen to plasmin. Plasmin activates matrix metalloproteases (MMPs) that destroy extracellular matrix components [12], degrades fibrin, and modulates the classical complement pathway, inducing cell migration and invasion, vasodilation, fibrinolysis, opsonization, and phagocytosis of pathogens [5]. The demonstration of activation of intranuclear mechanisms through the NFkB system and the ability to act on the Jak/Stat system involved in adhesion, migration, activation, proliferation, and survival, determines a synthesis of the elements of the cytokine system, TNF, IL-1β, IL-6, MMP-3, MCP-1, and NOS. Additionally, interaction with co-receptors, such as integrins and vitronectins, boosts this recognition (uPAR ligands) [13].

suPAR levels have been measured using different types of assays including enzyme-linked immunosorbent assays (ELISA), such as the CHORUS suPAR EXTENDED (Diesse Diagnostica Senese SpA, Siena, Italy), the Human uPAR Quantikine ELISA (R&D Systems, Minneapolis, MN, USA) and the suPARnostic ELISA (Virogates, Copenhagen, Denmark), and proteomics platforms, such as the aptamer-based assay SomaLogic SOMAscan (SomaLogic, Boulder, CO, USA) and the proximity extension assay Olink Explore (Olink, Uppsala, Sweden). suPAR is detectable in low, but constant, concentrations in the plasma of healthy individuals [14,15]. In the general population, suPAR levels are higher in females than in males [16]. Moreover, suPAR levels have been observed to increase with age [17]. Currently, suPAR is considered one of the strongest biomarkers of innate immunity activation, as its concentration positively correlates to the activation level of the immune system. Mirroring immune system activation, suPAR has been associated with many autoimmune rheumatic diseases [1,18], enabling the follow-up of disease activity, prediction of treatment response, or disease relapse [2]. Furthermore, suPAR levels positively correlate with pro-inflammatory biomarkers, including tumor necrosis factor-α (TNFα), C-reactive protein (CRP), and other parameters, including leukocyte counts. In contrast to CRP, suPAR levels in healthy individuals are stable throughout the day, regardless of whether the subject is fasting or not. suPAR concentrations are unaffected by repeated freeze-thaw procedures on plasma samples [19].

Thus, suPAR is a promising candidate as a clinical marker due to its high stability in plasma samples. In this review, we aimed to summarize the existing evidence of suPAR as a promising biomarker in different autoimmune rheumatic and non-rheumatic diseases. Specifically, the association of suPAR with Rheumatoid Arthritis (RA), Systemic Lupus Erythematosus (SLE), Systemic sclerosis (SSc), Behcet’s syndrome, Psoriasis, Ankylosing Spondylitis (AS), ANCA-Associated Vasculitis (AAV), Diabetes Mellitus type-1 (T1DM), Inflammatory Bowel Diseases (IBD), and Myasthenia Gravis (MG) is discussed.

## 2. Methods

The scientific literature involving suPAR in autoimmune diseases has been reviewed thoroughly. The research was carried out using PubMed and EMBASE databases, searching the following keywords: soluble urokinase receptor, Rheumatoid Arthritis, Ankylosing Spondylitis, Psoriasis, Psoriatic Arthritis, Juvenile idiopathic arthritis, Adult-Onset Still’s Disease, Anti-neutrophil cytoplasmic antibodies-associated vasculitis, Granulomatosis with polyangiitis, Microscopic polyangiitis, Eosinophilic granulomatosis with polyangiitis, Systemic Lupus Erythematosus, Idiopathic Inflammatory Myopathy, Polymyositis, Dermatomyositis, Immune-mediated necrotizing myopathy, Anti-synthetase syndrome, Sporadic inclusion body myositis, Sjögren’s Syndrome, Systemic Sclerosis, Inflammatory Bowel Diseases, Crohn’s disease, Ulcerative Colitis, Myasthenia Gravis, Multiple Sclerosis, Chronic Inflammatory Demyelinating Polyneuropathy, Graves’ Orbitopathy, Autoimmune Bullous Diseases, Uveitis, Behcet’s syndrome, Diabetes Mellitus. Only English-language articles were selected. Abstracts without the main text were excluded. The manuscript has been structured as a narrative review.

## 3. suPAR in Autoimmune Rheumatic Diseases

### 3.1. Rheumatoid Arthritis (RA)

RA is a chronic inflammatory disease characterized by leukocyte infiltration in synovial fluid and the surrounding synovial tissue [20,21]. However, various cells of the myeloid and leukocyte lineage, including monocytes/macrophages, neutrophils, B lymphocytes, mast cells, and subsets of T helper cells, mediate the intrinsic chain of events. Leukocytes are recruited to the inflamed area by chemokines and other chemo-attractants. Numerous growth factors, cytokines, and chemokines interact to regulate joint physiology, causing cartilage and bone degeneration. In 2010, Pliyev et al. conducted a comparative analysis of neutrophils isolated from the paired samples of synovial fluid (SF) and peripheral blood (PB) of RA patients. The blood concentration of suPAR was found to be significantly lower than in SF. Neutrophils are the predominant leucocytes accumulating in SF during the acute RA phase and represent the major cellular source of suPAR in SF. The authors demonstrated that SF neutrophils release the chemotactically active cleaved D2-D3 form of suPAR, which results in a prolonged inflammatory response [22]. Furthermore, Pliyev et al. [22] objectified the association of suPAR with the number of swollen joints [22]. The link between suPAR and disease activity in early RA was examined in a study by Enocsson et al. (2021). The authors evaluated suPAR levels at the disease onset and after 3 and 36 months in 252 Swedish patients affected by early RA. The authors demonstrated that suPAR levels were higher in RA patients at all-time points (baseline, 3 and 36 months) compared to healthy controls. Moreover, baseline suPAR levels were significantly associated with baseline disease activity, whereas suPAR levels correlated with joint damage (Larsen score > 5) at 36 months [23]. Recent research has already shown that suPAR is much more effective than CRP and ESR in identifying low-grade inflammation in RA. Toldi et al. analyzed the association of suPAR levels with DAS28 scores in RA patients. It emerged that suPAR levels in the subgroup with DAS28 < 2.6 (disease remission) were lower than in the subgroup with DAS28 > 2.6 (active disease) but still higher than in healthy controls (HC). The authors further demonstrated that suPAR values were significantly higher in patients with 4 tender and/or swollen joints than in patients with 2–3 or 0–1 tender and/or swollen joints. SuPAR may therefore be especially helpful in identifying inflammatory activity in patients who have clinical symptoms affecting joints but are in remission according to DAS28 scores [24]. Slot et al. evaluated the importance of suPAR in RA and other autoimmune diseases such as Reactive Arthritis (ReA) and Primary Sjogren’s Syndrome (pSS). The median suPAR value in RA was 1.47 μg/L (range 0.65–6.62), 0.68 μg/L in ReA (range 0.52–1.48), and 1.12 μg/L in pSS (range 0.76–1.92). SuPAR in RA also showed a positive correlation with CRP, ESR, and the number of swollen joints. The ReA group, on the other hand, had the lowest plasma suPAR concentrations while also having the highest CRP values of any group. The authors hypothesized that the higher suPAR concentrations in RA compared to ReA may be due to the relationship between suPAR and erosive activity [25].

Overall, these studies show that the uPA/uPAR interaction, in conjunction with the surrounding microenvironment and extracellular signals from various cell types, mediates RA pathogenesis. However, its role is in its early stages and will require further investigation in the near future.

### 3.2. Systemic Lupus Erythematosus (SLE)

SLE is a potentially severe autoimmune condition characterized by inflammation with a fluctuating clinical course of flares that may lead to permanent organ damage and quiescent periods, and it is associated with reduced quality of life, increased mortality, and accumulating irreversible organ damage [26].

Since the current information provided by CRP and ESR is constrained by their low sensitivity and low specificity, as well as by a number of preanalytical factors that can interfere with results, such as the diurnal cycle, sampling method, or even physical activity, there is a critical need to identify novel inflammation’s parameters [27].

Various studies have considered suPAR as a predictor of disease activity and organ damage in SLE. Enocsson et al. found that levels of suPAR were significantly higher in a group of 198 SLE patients compared to HC; however, no significant association with disease activity, neither defined as SLE disease activity index-2000 (SLEDAI-2K) nor as the physician’s global assessment (PGA), was found. Moreover, they found that the levels appeared to reflect irreversible organ damage, especially in the renal, ocular, and neuropsychiatric domains of the Systemic Lupus International Collaborating Clinics/American College of Rheumatology Damage Index (SDI). Of all the organ systems considered, renal damage had the most pronounced impact on suPAR levels [28].

In Toldi et al.’s study, 89 SLE patients were enrolled at various stages of disease duration and activity. suPAR plasma levels were elevated in SLE patients compared to HC patients and were revealed to be higher in SLE patients with vasculitis than in patients without vasculitis. Interestingly, in contrast with CRP and ESR, suPAR enabled the differentiation between patients with high disease activity (SLEDAI > 8) and those with moderate disease activity or in remission (SLEDAI ≤ 8) [29].

Another study conducted by Enocsson et al. in 2019 on 344 SLE patients demonstrated that suPAR levels at baseline were associated with global organ damage after 5 years of follow-up. In line with previous observations [28], there was no correlation with disease activity (SLEDAI-2K) at baseline, and no association with the presence of autoantibodies included in the ‘immunological disorder criterion’ of the ACR classification (of which anti-dsDNA antibody levels often parallel the disease activity). suPAR serum concentrations were found to correlate with musculoskeletal damage in SLE patients, which was responsible for the second-most frequent type of organ damage in the study cohort [30]. This is in line with the observations of Pliyev et al. [22] in a RA cohort objectifying the association of suPAR with the number of swollen joints, originating from synovial neutrophils, thereby recruiting leukocytes in the joints and promoting joint inflammation and subsequent damage.

Up to 60% of SLE patients have renal involvement, and lupus nephritis continues to be the leading source of morbidity and mortality over the disease’s progression. In the cohort of 202 patients with renal biopsy-proven lupus nephritis studied by Qin et al., active lupus nephritis had suPAR levels that were considerably greater than those of the SLE group without renal involvement and of healthy controls. Moreover, the patients with nephrotic syndrome, non-infectious leukocyturia, acute renal failure, and higher SLEDAI scores presented significantly higher baseline suPAR levels [31]. In addition, suPAR levels decreased significantly during remission, and the level of suPAR was found to be a risk factor for long-term renal outcomes [31].

Finally, in children with lupus nephritis, the increased amount of circulating suPAR has been associated with both multi-organ involvement and systemic inflammation [32].

SuPAR’s role in the pathogenesis of SLE is unknown, but its effects on leukocyte recruitment, phagocytic uptake of dying cells (efferocytosis), and complement regulation suggest that suPAR may have a central role in the pathogenesis of the disease.

### 3.3. Systemic Sclerosis (SSc)

Several biomarkers reflecting disease activity and severity in SSc have been described, but their role has not been completely clarified [33,34].

In a study by Legany et al. published in 2015, high levels of suPAR were found in SSc patients compared with HC patients, and higher levels were found in patients with anti-Scl-70 autoantibodies. Higher suPAR levels were found in patients with a diffuse cutaneous sclerosis variant compared with a limited cutaneous variant. suPAR levels also correlated with pulmonary involvement. suPAR levels correlated with a severe decrease in Diffusing Lung Capacity for Carbon monOxide (DLCO) and Forced Vital Capacity (FVC) values and increased levels were found in pulmonary fibrosis and pulmonary arterial hypertension within interstitial lung diseases. Higher suPAR levels were associated with vascular abnormalities (digital ulcers, Raynaud phenomenon, and pathological nailfold capillaroscopy) and with joint manifestations suggesting that this biomarker could become very important for early diagnosis and assessment of disease severity [35].

The association between suPAR levels and the severity of pulmonary involvement and the inverse correlation with DLCO in a recent study on 121 SSc patients was not confirmed. However, its levels were associated with the extent of skin involvement, autoantibody positivity, and vascular features (the presence of pulmonary arterial hypertension, telangiectasias, and present or past history of digital ulcers) [36].

### 3.4. Behcet’s Syndrome

Behcet’s syndrome is a chronic relapsing vasculitis characterized by multiple manifestations such as mucocutaneous lesions (oral and genital ulcers, erythema nodosum, and papulopustular lesions), arthritis, uveitis, neurological, gastrointestinal, and vessels involvements [37]. A study published by Saylam Kurtipek et al. in 2016 showed a statistically significant increase in suPAR levels in 30 patients with Behcet’s syndrome compared to 41 HC. They found a statistically significant difference between the two groups for CRP and suPAR levels but not for suPAR levels related to disease activity, even if the study has been conducted in a low number of patients and with early treatment with colchicine [38].

### 3.5. Psoriasis

Psoriasis is a chronic, immune-mediated inflammatory skin condition that has been associated with cardiovascular, metabolic, psychiatric, and arthritic comorbidities [39]. In a 2015 study, suPAR levels were analyzed in 65 Psoriasis patients versus a 50 HC cohort. There were no statistically significant differences in suPAR levels between the two groups [40]. These findings were confirmed in a more recent study. Moreover, suPAR levels were not associated with Psoriasis Area and Severity Index (PASI) scores, duration of disease, type of Psoriasis, treatment performed, and presence of arthralgias [41]. Both studies showed no utility of suPAR as a biomarker for Psoriasis, but studies in cohorts of patients with severe disease characterized by higher PASI scores have not yet been conducted.

### 3.6. Ankylosing Spondylitis (AS)

AS is an immune-mediated rheumatic disease characterized by chronic inflammation. The axial and sacroiliac joints are the main targets of the autoimmune reaction in AS. Spondylitis, a persistent inflammation of the spine, eventually results in the fusing of the vertebrae as the disease advances and extra bone is formed (ankylosis). Systemic autoimmune reactions can cause significant target organ damage in the later stages of the disease, causing enthesitis, peripheral arthritis, and extra-articular manifestations such as the gastrointestinal tract, eye inflammation, and heart inflammation [42]. As a result, early and reliable detection and monitoring of inflammation are critical in the management of AS.

In a 2013 study by Toldi et al. in a cohort of 33 AS patients at various stages of disease duration and activity and 29 HC, the CRP and ESR values were higher in AS patients than in healthy individuals, while suPAR values were comparable. There was a correlation found between BASDAI scores and CRP and ESR values in AS patients, but not with suPAR levels. So, these results do not support its usefulness in the assessment of inflammation in AS. For this reason, plasma suPAR levels are not used for monitoring the disease activity in AS such as ESR and CPR values [43].

### 3.7. ANCA-Associated Vasculitis (AAV)

AAV is characterized by pauci-immune necrotizing inflammation of the small blood vessels. AAV involves multiple organ systems including Granulomatosis with PolyAngiitis (GPA), Microscopic PolyAngiitis (MPA), and Eosinophilic Granulomatosis with Polyangitis (EGPA). The disease frequently presents as a pauci-immune focal segmental necrotizing glomerulonephritis with a very rapid decline in renal function. The kidney is the most organ susceptible to the disease. MyeloPeroxidase (MPO) and PRoteinase 3 (PR3) are the two most significant target antigens of ANCAs, which are the serologic markers of AAV [44]. SuPAR is linked to kidney disorders and has been used to predict the progression of renal function loss. The objective of Fei Huang et al.’s study was to determine whether suPAR levels were associated with ANCA-AAV disease activity. SuPAR levels in 90 AAV patients were assessed, and it was found that they were significantly higher than in HC patients. Additionally, suPAR was found to be significantly higher in AAV patients who were in active stages of the disease compared to those who were in partial remission, indicating a link between disease activity and a poor prognosis [45]. Similar findings were found in another study by Rowaiye et al. where 60 patients with renal involvement and AAV have been followed up for a median period of 12 months with higher levels of suPAR in patients with relapse compared with patients with inactive AAV. These findings suggest that plasma suPAR levels are a predictive biomarker for renal AAV patients’ mortality risk categorization. Notably, higher baseline suPAR levels were found in non-survivors than in survivors. In 2021, Zabinska et al. observed a statistically significant correlation between suPAR levels and kidney function in AAV with renal involvement associated with the immune activation state. So, the observed correlation between suPAR and the reported clinical characteristics may suggest that suPAR has the capacity to indicate renal involvement and disease severity [44].

### 3.8. Key Messages

In summary, the body of research described in this review provides support for the hypothesis that suPAR can be considered a novel and valuable biomarker in several autoimmune rheumatic diseases.

In RA, most of the studies suggest many advantages of suPAR in detecting low-grade inflammation and agree with the association of suPAR with the number of swollen joints [22] and disease activity [23,24].

Various studies have considered suPAR as a predictor of disease activity and organ damage in SLE [29]; however, in other research, no significant association with disease activity was found [28]. suPAR correlates with musculoskeletal damage in SLE [30], as well as autoantibody positivity, skin involvement, and vascular features [36]. SuPAR is able to discriminate between SLE patients with high disease activity and those with less active disease or those in remission [29]. Regarding Behcet’s syndrome, Psoriasis, and AS, the number of current studies is not yet sufficient to draw conclusions on the effectiveness of suPAR. On the contrary, in AAV, suPAR levels are valuable predictive biomarkers for renal involvement and disease severity [44].

## 4. suPAR in Autoimmune Non-Rheumatic Diseases

### 4.1. Type 1 Diabetes Mellitus (T1DM)

Endothelial dysfunction caused by chronic inflammation is the key event responsible for the development of micro- and macrovascular complications in Type 1 Diabetes Mellitus (T1DM) [46]. Since being considered a new marker of inflammation and endothelial dysfunction, suPAR has generated great scientific interest in patients with T1DM.

In 2014, Theilade et al. compared suPAR levels in HC and T1DM patients and then investigated the association of suPAR levels with disease duration and disease complications. This observational study demonstrated that suPAR concentrations were significantly lower in the control group (*n* = 51) compared to the group of T1DM patients (*n* = 667). However, T1DM patients with normoalbuminuria and short disease duration had levels that were comparable to those of control subjects. Elevated suPAR levels in diabetics were significantly correlated with the length of T1DM (>10 years) and the severity of albuminuria. Regardless of other risk factors such as cardiovascular disease, albuminuria, autonomic dysfunction, and a high level of arterial stiffness, suPAR was associated with disease complications. There is a quite good correlation between suPAR levels, the estimated glomerular filtration rate (eGFR), and urine albumin excretion rate. The elevated levels of suPAR in T1DM are likely caused by complications rather than diabetes itself. Additionally, it was discovered that suPAR levels rose gradually in patients who had more disease complications, suggesting that suPAR may be a potentially helpful marker of disease severity in T1DM [47].

In another 2016 study, Theilade et al. investigated whether higher levels of suPAR were associated with subclinical impaired cardiac function assessed with tissue doppler and speckle tracking echocardiography in T1DM patients without known heart disease or end-stage renal disease and with normal left ventricular ejection fraction. Moreover, it was demonstrated that suPAR levels were significantly associated with early systolic and diastolic myocardial impairment, assessed using sensitive echocardiographic methods. From these observations, there is strong evidence suggesting that suPAR is a sensitive marker of early and discrete myocardial dysfunction [48].

### 4.2. Inflammatory Bowel Diseases (IBD)

IBD, which includes Crohn’s disease (CD) and ulcerative colitis (UC), are gastrointestinal tract inflammatory chronic diseases that can affect people at any age. Currently, Fecal Calprotectin (FC) is the most important and useful non-invasive marker for intestinal inflammation both in adults [49] and children [50]. However, FC is limited by great intra-individual variability during a single day [51] and by low compliance, especially in the pediatric age. Among traditional biomarkers, ESR and CRP are not specific for intestinal inflammation and may be elevated in many other situations such as infections or other inflammatory conditions. Furthermore, CRP may be low in 10 to 20% of pediatric patients with severe UC [52] and therefore a new blood biomarker to monitor disease activity in pediatric IBD patients is required. Kolho et al. tested suPAR levels in IBD pediatric patients (CD *n* = 19, UC *n* = 16) at the beginning of therapy with glucocorticoids or TNF-antagonist. In both groups, suPAR levels were low and only in the patients treated with glucocorticoids, there was a slight decline in suPAR levels after the introduction of therapy. The study did not reveal any correlation between suPAR levels and clinical disease activity, neither with ESR nor with CRP levels. Consequently, the authors concluded that suPAR is of limited value in assessing systemic inflammatory response in pediatric IBD [53]. Nevertheless, future studies on the role of suPAR as a possible non-invasive marker of IBD in pediatric care are needed.

### 4.3. Myasthenia Gravis (MG)

The neuromuscular junction autoimmune illness MG is characterized by cyclical muscle weakening. At the neuromuscular junction, complement and anti-muscle nicotinic acetylcholine receptor (AChR) antibodies are essential for the inflammation and the death of motor endplates [54]. To predict clinical outcomes, a biomarker indicating MG disease activity is desirable, but none has yet been discovered. In a study by Uzawa et al., the potential role of suPAR was evaluated as a biomarker of MG disease activity in 40 patients with AChR antibody-positive MG and in 30 HC, and its correlation with clinical variables and severity scale scores was investigated. suPAR levels significantly correlated with MG activities of daily living scale and MG Foundation of America classification at serum sampling, but not with AChR antibody titers. Despite several limitations of the study, such as the sample size and lack of data, the authors concluded that suPAR levels could be a potential candidate as a novel biomarker of disease activity in MG patients with AChR antibody positivity, as they may reflect the severity of MG-related neuromuscular junction damage [55]. suPAR levels significantly correlated with MG severity scale scores, due to the possibility that high suPAR levels up-regulate plasmin production, which in turn causes inflammation and harm to the neuromuscular junction via the modulation of complement and immune cells, such as macrophages.

### 4.4. Key Messages

In summary, the body of evidence described in this review suggests the ability of suPAR to be a potential biomarker also in autoimmune non-rheumatic diseases.

Regarding T1DM, suPAR has been mainly linked to the duration of the disease, the degree of albuminuria, and disease severity [47].

Concerning IBD, current studies suggest that suPAR is of limited value in assessing the systemic inflammatory response [53].

The activity of suPARas a biomarker of disease activity in MG has yet to be established. However, suPAR levels can be a candidate as an innovative biomarker of disease activity in AChR antibody-positive MG [55].

## 5. Conclusions and Perspective

Our review provides a promising role of suPAR as an additional prognostic biomarker of systemic inflammation. In Figure 2, we resume the key information on the existing evidence of suPAR as a promising biomarker in different autoimmune rheumatic and non-rheumatic diseases. In contrast to CRP, primarily synthesized by hepatocytes in response to an increased immune activation, suPAR is upregulated and released to the bloodstream by innate immune cells, [1,5]. suPAR serum levels, therefore, systemically mirror ongoing local inflammation [2]. suPAR has inflammatory functions and is mostly positively correlated with established inflammatory biomarkers, including CRP [2]. Obviously, none of the biomarkers is particularly specific to any inflammatory disease, but they have shown to be elevated and to provide strong prognostic value in many different autoimmune rheumatic and non-rheumatic disorders. As such, suPAR is probably of limited value as a diagnostic tool, but it can contribute by adding information about the systemic chronic inflammatory state. The significant difference in the half-life of biomarkers (e.g., 19 h for CRP and >7–10 days for suPAR) [56] attributes multiple aspects of the clinical information. Therefore, even though suPAR is positively correlated with the recognized markers of inflammation, the correlation with many of them is weak, particularly with CRP.

In contrast to IL-6 [57], Tumor Necrosis Factor alfa (TNF-α), and other cytokines, suPAR is relatively stable in the blood, not subjected to diurnal variation, and readily quantifiable in both healthy and diseased individuals by means of routinely applicable analytical assays [2]. In addition, suPAR possesses important features to serve as a suitable clinical prognostic biomarker: the protein reflects (i) ongoing inflammation, enabling the prediction of incident or prevalent disease, (ii) the extent or severity of the disease, (iii) the risk of relapse, and (iv) the response to anti-inflammatory intervention or disease remission [2]. In particular, higher suPAR concentrations are found in patients with chronic inflammatory rheumatic diseases such as SLE, SSc, RA, and Behcet’s syndrome. suPAR correlates with renal involvement and disease activity in SLE and AAV. Furthermore, as previously reported, suPAR correlates with a musculoskeletal domain in SLE and with erosive damage in RA. For these reasons, it can identify patients who need an accurate and tight follow-up. In addition, the clinical usefulness of suPAR as a marker of disease activity has been shown not only in rheumatic diseases but also in other autoimmune diseases such as T1DM where it correlates with renal involvement and early myocardial dysfunction and MG where it is associated with neuromuscular junction damage.

The prognostic value of suPAR could have important therapeutic consequences. Similarly to COVID-19, where elevated suPAR levels first identified patients at risk of progressing to severe respiratory failure or death that needed to initiate early targeted treatment with anakinra, patients with autoimmune diseases identified with higher suPAR levels might receive earlier treatment of the inflammatory activity and, in case of unsatisfactory decline in response to treatment initiation, an earlier switch to another, hopefully more effective, therapeutic regime [58].

In conclusion, serum levels of suPAR represent a valuable prognostic biomarker of autoimmune rheumatic and non-rheumatic diseases, by systemically reflecting the local ongoing inflammation, thereby potentially guiding clinicians in the therapeutic decision-making process.

## Figures and Tables

**Figure 1 jpm-13-00688-f001:**
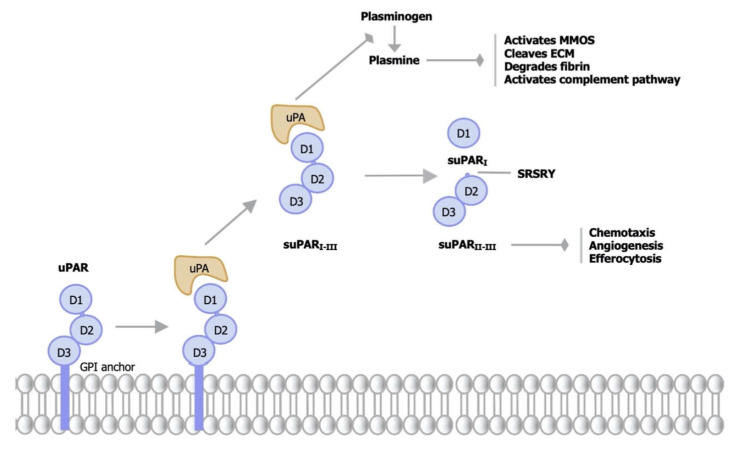
Structure of uPAR and suPAR isoforms. uPAR is bound to the membrane by a glycosyl posphatidylinositol (GPI) anchor and suPAR is its soluble form. The protein consists of three domains, D1–D3, connected with a linker region between D1 and D2–D3. Binding of urokinase plasminogen activator (uPA) releases suPAR I-III. Active uPA cleaves plasminogen to plasmin that activates matrix metalloproteases (MMPs), cleaves Extracellular Matrix Components (ECM), degrades fibrin, and activates the complement pathway. Subsequent proteolytic cleavage in the linker region of suPAR II-III, reveals in the suPAR I and suPAR II-III isoforms, exposing an SRSRY sequence that promotes chemotaxis, stimulates angiogenesis, and inhibits neutrophil efferocytosis.

**Figure 2 jpm-13-00688-f002:**
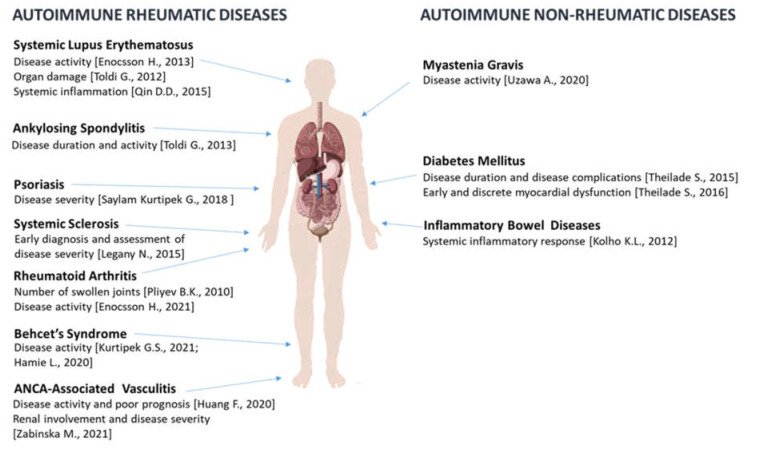
Overview of autoimmune rheumatic and non-rheumatic diseases with elevated suPAR levels. According to clinical studies, high suPAR levels have been associated with disease activity, severity, and prognosis in many autoimmune rheumatic diseases, including SLE [28,29,31], AS [43], Psoriasis [38], SSc [35], RA [22], Behcet’s syndrome [40,41], and AAV [44,45] as well as in non-rheumatic diseases including MG [55], T1DM [47,48], and IBD [53].

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
