# Peer review of "Soluble Urokinase Plasminogen Activator Receptor (suPAR) in Autoimmune Rheumatic and Non Rheumatic Diseases"

_jpm, 2023, doi:10.3390/jpm13040688_

Round 1
Reviewer 1 Report
The article discusses the potential role of suPAR as a biomarker in various autoimmune rheumatic and non-rheumatic diseases. It starts with a biological background of suPAR followed by a brief method for literature research. Then, suPAR in either autoimmune rheumatic disease or autoimmune non-rheumatic diseases was discussed. In general, the manuscript is well organized and written. However, I believe that the quality of the review can be improved after a major revision.
Comments:
1. A review usually covers a plenty of research works, and normally cites at least hundred publications. However, only 55 articles are cited in the present review. Are all relevant publications are discussed in this review?
2. It is suggested to include Figures in section 3 and section 4 to help readers grab the key information better.
3. Critical comments are missing from section 3 and section 4.
4. The article highlights that suPAR is elevated in various disease, and can be used as a potential marker. However, readers can hardly get this point from the title. Besides, it would be better to supply some information about the determination of suPAR.
5. A perspective is suggested to be included at the end of the article.
Other comments:
1. Introduction, line 4: [1], [2], [3], [4] or [1-4]? The authors should double-check the reference style thoroughly to ensure it fits JPM well.
2. Typing errors: Instead of “autoimmune rheumatic ad non- rheumatic diseases“, it should be “autoimmune rheumatic and non- rheumatic diseases“ (The last paragraph in Introduction, Line 3); the extra comma should be removed in the last sentence. Such kind of errors should be avoided in the revised manuscript.
3. The authors are highly recommended to carefully check the whole article, such as typing and layout.
Author Response
The article discusses the potential role of suPAR as a biomarker in various autoimmune rheumatic and non-rheumatic diseases. It starts with a biological background of suPAR followed by a brief method for literature research. Then, suPAR in either autoimmune rheumatic disease or autoimmune non-rheumatic diseases was discussed. In general, the manuscript is well organized and written. However, I believe that the quality of the review can be improved after a major revision.
Comments:
- A review usually covers a plenty of research works, and normally cites at least hundred publications. However, only 55 articles are cited in the present review. Are all relevant publications are discussed in this review?
We thank the reviewer for the comment. It is important to notice that the literature regarding suPAR as a biomarker is very recent, with regard to autoimmune diseases. We accurately searched the relevant publications on this topic and we discussed it.
- It is suggested to include Figures in section 3 and section 4 to help readers grab the key information better.
We included Figure 2 in the text (see section 5), it resumes the key information of the existing evidences of suPAR as promising biomarker in different autoimmune rheumatic and non rheumatic diseases.
- Critical comments are missing from section 3 and section 4.
We included a critical comment at the end of both sections as new paragraphs called “Key messages” to summarize and critically discuss the scientific evidences.
- The article highlights that suPAR is elevated in various disease, and can be used as a potential marker. However, readers can hardly get this point from the title. Besides, it would be better to supply some information about the determination of suPAR.
We changed the title into: “Soluble urokinase Plasminogen Activator Receptor (suPAR) as a biomarker of autoimmune diseases” according your suggestion. We also added information about the determination of suPAR levels in the 1. introduction section.
- A perspective is suggested to be included at the end of the article.
We included a perspective at the end of the review (see section 5 Conclusion and Perspective) to help readers to understand the possible clinical application of the biomarker in the next future.
Other comments:
- Introduction, line 4: [1], [2], [3], [4] or [1-4]? The authors should double-check the reference style thoroughly to ensure it fits JPM well.
In agreement with the reviewer, we corrected the reference style.
- Typing errors: Instead of “autoimmune rheumatic ad non- rheumatic diseases“, it should be “autoimmune rheumatic and non- rheumatic diseases“ (The last paragraph in Introduction, Line 3); the extra comma should be removed in the last sentence. Such kind of errors should be avoided in the revised manuscript.
We corrected the typing errors in the text.
- The authors are highly recommended to carefully check the whole article, such as typing and layout.
We corrected the typing errors in the text and reviewed the layout.
Reviewer 2 Report
In the current paper, Manfredi et al. reviewed literature that reported the changes in soluble urokinase-type plasminogen activator receptor (suPAR) levels in autoimmune rheumatic diseases such as rheumatoid arthritis, systemic lupus erythematosus, systemic sclerosis, and Behcet's syndrome ad non- rheumatic diseases such as diabetes mellitus, inflammatory bowel diseases and myasthenia gravis. They attempt to find a correlation between the suPAR level, mainly in blood or other body fluid, and the disease-associated symptoms, activity, prognosis, or tissue damage.
Manfredi et al., summarized previous studies and used relevant references in the field with minimum but necessary self-citation. The conclusion was parallel with the review's main text. They used figure 1 to illustrate the structure of uPAR and suPAR isoforms. Overall, this manuscript showed a comprehensive review and is worth publication.
Author Response
In the current paper, Manfredi et al. reviewed literature that reported the changes in soluble urokinase-type plasminogen activator receptor (suPAR) levels in autoimmune rheumatic diseases such as rheumatoid arthritis, systemic lupus erythematosus, systemic sclerosis, and Behcet's syndrome ad non- rheumatic diseases such as diabetes mellitus, inflammatory bowel diseases and myasthenia gravis. They attempt to find a correlation between the suPAR level, mainly in blood or other body fluid, and the disease-associated symptoms, activity, prognosis, or tissue damage.
Manfredi et al., summarized previous studies and used relevant references in the field with minimum but necessary self-citation. The conclusion was parallel with the review's main text. They used figure 1 to illustrate the structure of uPAR and suPAR isoforms. Overall, this manuscript showed a comprehensive review and is worth publication.
We thank the reviewer for the positive comment.
Round 2
Reviewer 1 Report
All problems are addressed. I would recommend the manuscript for publication in JPM.